

# Experimental modulation of external microbiome affects nestmate recognition in harvester ants (*Pogonomyrmex barbatus*)

Andy Dosmann[1], Nassim Bahet[2,3] and Deborah M. Gordon[3]

[1] Department of Natural Sciences, Minerva Schools at Keck Graduate Institute, San Francisco, CA, United States
[2] Stanford Online High School, Stanford, CA, United States
[3] Department of Biology, Stanford University, Stanford, CA, United States

## ABSTRACT

Social insects use odors as cues for a variety of behavioral responses, including nestmate recognition. Past research on nestmate recognition indicates cuticular hydrocarbons are important nestmate discriminators for social insects, but other factors are likely to contribute to colony-specific odors. Here we experimentally tested whether external microbes contribute to nestmate recognition in red harvester ants (*Pogonomyrmex barbatus*). We changed the external microbiome of ants through topical application of either antibiotics or microbial cultures. We then observed behavior of nestmates when treated ants were returned to the nest. Ants whose external microbiome was augmented with microbial cultures were much more likely to be rejected than controls, but ants treated with antibiotics were not. This result is consistent with the possibility that external microbes are used for nestmate recognition.

Corresponding author
Andy Dosmann,
dosmann1@gmail.com

## INTRODUCTION

In ants, individuals discriminate between nestmates and non-nestmates, which is often equivalent to kin discrimination, by using chemical cues (*Vander Meer & Morel, 1998*; *Van Zweden & D'Ettorre, 2010*; *Sturgis & Gordon, 2012*). Cuticular hydrocarbons (CHCs) are chemical compounds that serve a variety of functions in social insects but very often are implicated in nestmate recognition by both empirical and theoretical research (*Wagner et al., 2000*; *Howard & Blomquist, 2005*). CHCs are synthesized from lipids, and the CHC profile of an ant, which consists of various classes of alkanes, alkenes and methyl-branched hydrocarbons in differing proportions, can be both task and colony specific (*Sturgis & Gordon, 2012*). In *Cataglyphis niger*, when hydrocarbons are isolated from the postpharyngeal gland and applied to one of a pair of live ants, the untreated ant behaves in a way that indicates CHCs function as nestmate discriminators (*Lahav et al., 1999*). *Esponda & Gordon (2015)* recently proposed a model of distributed nestmate recognition in which each ant in a colony has a flexible decision boundary in multidimensional odor space, acquired through interactions, that divides nestmates from non-nestmates.

For good reason, research on nestmate recognition in social insects has focused on CHCs, but other chemicals are also likely to play a role in nestmate recognition (*Sturgis & Gordon, 2012*). *Davis et al. (2013)* reviewed studies showing that volatile chemicals emitted by microbes affect insect behavior ranging from foraging to oviposition. *Lizé, McKay & Lewis (2014)* showed that bacteria influence kin recognition in *Drosophila melanogaster*. Although CHCs alone are sufficient for nestmate recognition in some species of ants (*Lahav et al., 1999*; *Wagner et al., 2000*), this does not preclude a role for microbes. Both internal and external microbial communities are colony-specific in ants (*Anderson et al., 2013*; *Hu et al., 2014*), and thus potential nestmate discriminators. Over ten years ago, *Matsuura (2001)* experimentally showed that bacteria mediate nestmate recognition in termites (*Reticulitermes speratus*). However, the role of microbes in nestmate recognition of ants has not been investigated, except for a study by *Richard et al. (2007)* showing that ants from one colony of fungus growing ants (*Acromyrmex spp.*), if reared on a second colony's fungus, experienced less aggression when introduced to the second colony.

Here we test the hypothesis that the microbial composition on an ant's cuticle affects whether it is treated as a nestmate. Both pathogenic and protective microbes inhabit the surface of insect cuticles (*Douglas, 2015*), and they could produce chemicals that affect nestmate recognition (*Davis et al., 2013*). Previous work in honey bees (*Apis mellifera*) showed that internal injection of bacteria caused significant changes in CHC profiles, and in turn nestmate recognition (*Richard, Holt & Grozinger, 2012*), but here we isolate the effect of external microbes on nestmate recognition. To test the hypothesis that the external microbiome affects nestmate recognition in ants, we experimentally modulated the microbes on the cuticles of red harvester ants (*Pogonomyrmex barbatus*) and tested whether those ants were rejected by their nestmates. *P. barbatus* can distinguish between nestmates and non-nestmates using CHCs alone (*Wagner et al., 2000*), but microbes may still play a role. We predicted that if the external microbiome plays a role in nestmate recognition, individuals with an altered external microbiome would not be accepted as nestmates when returned to their nest.

## METHODS

### Study species

Experiments were conducted in 3 queenright, brood-producing laboratory colonies of red harvester ants (*Pogonomyrmex barbatus*) obtained from a field site near Rodeo, NM. Details of lab conditions are found in *Gordon et al. (2005)*.

### Assay of nestmate recognition

On the day of the experiment, we first removed a group of ants from their nest's foraging arena ∼4 h prior to behavioral testing. Group size (9–20 ants) depended on the number of ants active in the foraging arena during sampling. Each ant then received one of five experimental or control treatments, described below, and was painted with a color corresponding to its treatment for identification during behavioral observation (*Gordon et al., 2005*). We then returned the group of treated and marked ants to a location near

the nest entrance, where they interacted with resident nestmates. We performed the test with groups of ants at the nest entrance, rather than paired individuals in a foreign arena, because recent models indicate that accurate nestmate recognition occurs at the colony level (*Johnson, Van Wilgenburg & Tsutsui, 2011*; *Esponda & Gordon, 2015*) and empirical work indicates that testing ants in groups provides results that are more consistent and indicative of natural responses than tests with individuals (*Roulston, Buczkowski & Silverman, 2003*). An observer blind to the treatment groups then scored whether each ant was rejected from the nest during the following 20 min. Rejection, which was often preceded by aggressive acts like biting, consisted of a resident ant seizing a treated ant and carrying it out of the nest or away from the nest entrance towards the midden pile at the far end of the foraging arena.

### Experimental treatment

We manipulated the external microbiome of ants ($N = 111$) using five different treatments: (1) Antibiotic group: to decrease the external microbiome, we coated each ant's body surface with a 1% rifampin/sterile water solution. (2) Microbial group: to increase the external microbiome, we coated each ant's body surface with a solution of Lysogeny Broth (LB) broth (Becton Dickenson, Franklin Lakes, NJ, USA) that had previously been cultured from nestmates. Because only a small proportion of the microbiome survives in culture (*Rappe & Giovannoni, 2003*), this solution modified and augmented an ant's external microbiome, and we then confirmed this change (see below). In the control groups, we treated ants with (3) Water: sterile water, (4) Broth: sterile LB broth, or (5) Nest: simply removed ants from the nest and marked them before return.

### Validation of treatment

To validate that our treatment produced the intended effect on the ants' external microbiome, we used the method of *Ren et al. (2007)* to culture the external microbiome of additional ants ($N = 35$). Although only a small proportion of microbial species can be cultured (*Rappe & Giovannoni, 2003*), we used differences in the amount of culturable microbes obtained, measured by counting the number of microbial colony forming units (CFUs) that grew on LB agar plates, to indicate effectiveness of the treatments. A greater number of CFUs indicates a greater amount of external microbes on an ant.

### Effect of treatment on survival

To determine whether the treatments had any beneficial or adverse effects, we treated additional ants ($N = 72$) and kept them in Petri dishes with water for two weeks. We checked on individuals once daily and recorded whether the ant was alive or dead.

### Statistical analysis

To control for non-independence produced by colony membership and returning ants in groups (i.e., block design), we used a generalized linear mixed-effects model fitting block and colony as random effects, with treatment (Microbial, Antibiotic, LB Broth, Water, Nest) as a categorical predictor and rejection as a binary dependent variable. We used the *lmer* function in the *lme4* package in R 3.0.2 for linear mixed-effects models, and the *multcomp* package to make Tukey post hoc comparisons of treatment effects.

Because the counts of CFUs were nonparametrically distributed, we used Wilcoxon Rank Sum Tests to assess the treatment validation. We made planned comparisons that would indicate treatment efficacy (Antibiotic vs. Broth, Water, and Nest; Microbial vs. Broth, Water, and Nest), correcting alpha for the false discovery rate according to *Benjamini & Hochberg (1995)*.

To determine whether an experimental treatment significantly increased or decreased survival, we used a Cox proportional hazards model with the *survival* package in R (*Crawley, 2007*), and used the *multcomp* package to make Tukey post hoc comparisons of treatment effects.

## RESULTS

The experimental treatment modulated ants' external microbiome, as measured by number of colony forming units. The topical application of the microbial culture significantly increased in the number of CFUs obtained from an ant's cuticle compared to each of the controls (Microbial vs. Broth: $W = 49$, $P = 0.002$, $N = 10$, vs. Water: $W = 40.5$, $P = 0.006$, $N = 9$, vs. Nest: $W = 49$, $P = 0.002$, $N = 10$). Antibiotic treatment significantly decreased the number of CFUs compared to controls, except for the water treatment (Antibiotic vs. Broth: $W = 7.5$, $P = 0.035$, $N = 10$; vs. Water: $W = 13.5$, $P = 0.317$, $N = 9$; vs. Nest: $W = 7$, $P = 0.026$, $N = 10$). With 5 of 6 comparisons $P < 0.05$, our alpha, controlled for false discovery rate, is 0.042.

Changes in an ant's external microbiome influenced whether it was rejected from its nest by nestmates (Fig. 1). Ants with experimentally augmented external microbiomes were more likely to be rejected than ants treated with broth alone, sterile water, or only marked (all post hoc comparisons $|Z| > 3.8$, $P < 0.001$). Ants treated with the antibiotic solution were not more likely to be rejected than ants treated with broth alone, sterile water, or only marked (all post hoc comparisons $|Z| < 1.1$, $P > 0.80$).

Ants in the Microbial group died significantly sooner than ants given the water treatment ($Z = -3.085$, $P = 0.017$). There were no significant differences in survival between the Microbial group and the other control groups (all post hoc comparisons $|Z| < 1.64$, $P > 0.46$), or between the Antibiotic group and the control groups (all post hoc comparisons $|Z| < 1.173$, $P > 0.76$).

## DISCUSSION

We found that as we predicted, ants treated with additional microbes were rejected by resident colony members as if they were non-nestmates. This aligns with our predictions and supports our hypothesis that the external microbiome plays a role in nestmate recognition in *P. barbatus*. Contrary to predictions, ants treated with the antibiotic solution were not rejected by resident colony members. Because the antibiotic treatment was delivered in a sterile water solution, ants' CHC profiles were likely unaltered, in which case many familiar odors would persist. This is consistent with the possibility that the presence of a foreign odor is more likely to lead to the recognition of a non-nestmate than absence of a familiar odor (*Van Zweden & D'Ettorre, 2010*; *Esponda & Gordon, 2015*), as well as experimental

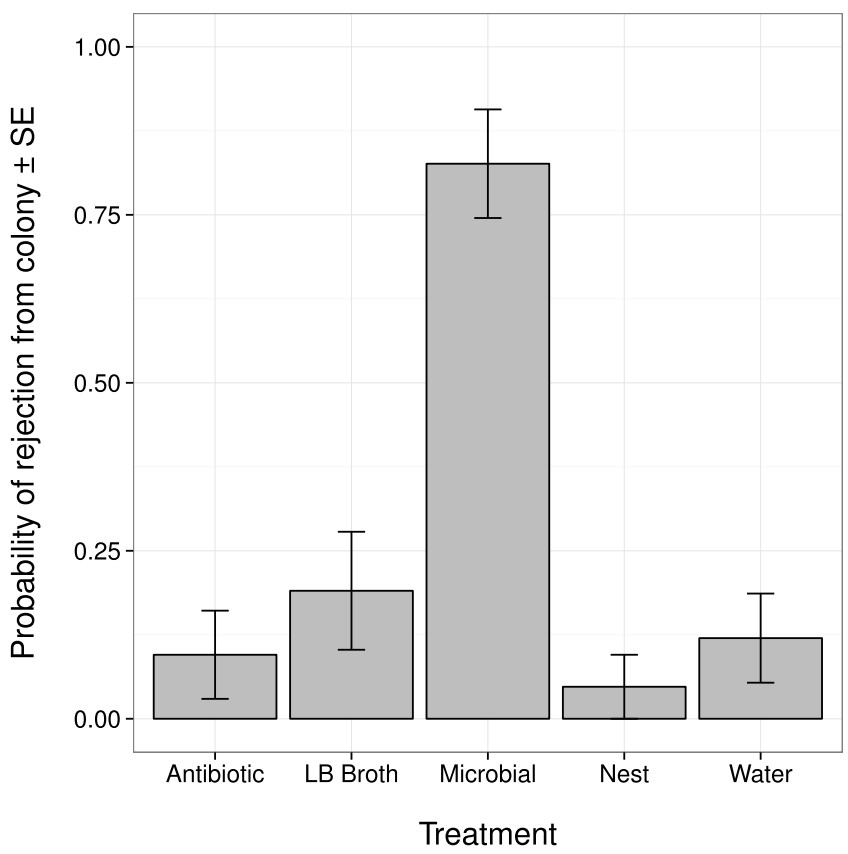

**Figure 1** **Probability that an ant was rejected by its nestmates (i.e., not recognized as a nestmate) after manipulation of its external microbiome.** Antibiotic, topical treatment with 1% rifampin solution. LB Broth, topical treatment with sterile LB broth. Microbial, topical treatment with microbes cultured from ants in LB broth. Nest, no topical treatment; individuals were only marked before return to nest. Water, topical treatment with sterile water.

data showing that carpenter ants (*Camponotus herculeanus*) reject non-nestmates by recognizing novel CHCs (*Guerrieri et al., 2009*).

The rejection of ants with augmented external microbiomes could be interpreted as a form of social immunity (*Cremer, Armitage & Schmid-Hempel, 2007*), in which ants with an unfamiliar or unusual microbiome are perceived as sick nestmates in need of quarantine. Social immunity may involve a microbially-mediated form of nestmate recognition, since a microbe that causes an individual to be placed on the 'reject' side of another ant's boundary may or may not be a pathogen. In this way, social immunity could constitute an additional function of microbially-mediated nestmate recognition. We found a significant difference in survival between ants from the Microbial group and those treated with water, but not between ants from the Microbial group and those from the Broth and Nest groups. Thus, the extent to which the microbial treatment may have included pathogens is not resolved and we do not know the extent to which rejection of treated ants contributed to social immunity.

In addition to further research on social immunity, our results suggest several research directions that may provide insight on the role of microbes in chemical communication.

First, it will be important to learn what microbial species are key to nestmate recognition and what chemicals they produce. In hyenas, fermentative bacteria appear to play a key role in chemical communication (*Theis et al., 2013*). Second, it will be important to establish how much microbial communities differ among colonies, and how consistent those communities are over time. In leafcutter ants (*Acromyrmex echinatior*), the external microbiome is colony-specific and differences are maintained up to 10 years, even after colonies were kept in laboratory conditions (*Anderson et al., 2013*). Third, an ant's microbiome may be task specific, as well as colony-specific. In *P. barbatus*, CHCs are task specific (*Sturgis & Gordon, 2013*) and influence task allocation (*Greene & Gordon, 2003*). It is likely that microbial communities of ants vary with task type, as foragers are exposed to different microbes from brood care or midden workers (*Cremer, Armitage & Schmid-Hempel, 2007*). Microbes may also inform task decisions alongside CHCs, or could directly affect the structure of CHCs.

In conclusion, our results support a role for microbes in nestmate recognition in harvester ants. They align with experimental data showing microbes influence nestmate/non-nestmate recognition in termites (*Matsuura, 2001*) and leaf-cutting ants (*Richard et al., 2007*), and show that external microbes alone can impact nestmate recognition. This finding augments extensive evidence of CHC-mediated nestmate recognition in social insects (*Vander Meer & Morel, 1998*; *Van Zweden & D'Ettorre, 2010*; *Sturgis & Gordon, 2012*), and calls for further investigation of the role of microbes in chemical communication of social insects.

### Funding

Stanford's Thinking Matters Program Fund and the CISCO Research Fund provided support for the study. The funders had no role in study design, data collection and analysis, decision to publish, or preparation of the manuscript.

### Grant Disclosures

The following grant information was disclosed by the authors:
Stanford's Thinking Matters Program Fund.
CISCO Research Fund.

### Competing Interests

The authors declare there are no competing interests.

### Author Contributions

- Andy Dosmann conceived and designed the experiments, performed the experiments, analyzed the data, contributed reagents/materials/analysis tools, wrote the paper, prepared figures and/or tables, reviewed drafts of the paper.
- Nassim Bahet conceived and designed the experiments, performed the experiments, reviewed drafts of the paper.
- Deborah M. Gordon conceived and designed the experiments, contributed reagents/materials/analysis tools, reviewed drafts of the paper.

## Data Availability

Data is available in the Supplemental Information.

## Supplemental Information

Supplemental information for this article can be found online at http://dx.doi.org/10.7717/peerj.1566#supplemental-information.

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
