# Peer review of "Experimental modulation of external microbiome affects nestmate recognition in harvester ants (Pogonomyrmex barbatus)"

_PeerJ, doi:10.7717/peerj.1566_

## Round 0.1 · original submission · Minor Revisions

Both reviewers and I agree that the paper is a nice, neat and well executed study. It is quite a short manuscript, so I believe that the authors can flesh out many of their points identified by both reviewers.

I am also not familiar with the Cox proportional hazards model under such circumstances. A more commonly used statistics to assess survival are generalized linear model. E.g. from
e.g. from Andrew N. R., Hart R. A., Jung M.-P., Hemmings Z. & Terblanche J. S. (2013) Can temperate insects take the heat? A case study of the physiological and behavioural responses in a common ant, Iridomyrmex purpureus (Formicidae), with potential climate change. J Ins Phys 59, 870-80.

'The proportion of ants surviving at each temperature was assessed
using a generalized linear model (GLZ) with a binomial
distribution and a logit-link function in R 2.14.1 (R Development
Core Team, 2011). Since survival is a binary variable (alive or
dead), we used a non-parametric generalized linear model approach
(with the default logit-link function of errors given that
the survival proportions are bounded at each end of the distribution),
to analyze the effects of temperature on survival. This approach
is similar in principal to a logistic regression (see
Crawley, 2007 for background on this method and e.g. in Terblanche
et al., 2008).'

Please confirm that Cox proportional hazards model is appropriate here - and justify with relevant references.

For your Figure 1 - I would also prefer a more informative box plot - or at least an average point +/- s.e..

·

Basic reporting

No Comments

Experimental design

No Comments

Validity of the findings

No Comments

Additional comments

The authors present a worthwhile study on the effects of the external microbial community on nestmate recognition in ants. Through experimental manipulation, they decrease or increase the microbial load of foraging ants, and record the subsequent rejection rate by nestmates at the nest entrance. Increase of microbial load through the application of a microbial ‘broth’ is found to increase rejection, while decrease of microbial load through the application of an antibiotic has no measurable effect.

The study is well-designed, with clearly stated research questions, and clean-cut results. The methods are appropriate for the research questions. The manuscript is written in a very succinct and clear style, and reads well. I believe the manuscript is fit for publication in PeerJ after minor revisions. My suggestions on how the manuscript could be further improved are outlined below.


L. 50 ff. Due to the demonstrated importance of CHCs in nestmate recognition, it would be nice to have a little bit more info here (prevalence, make-up, source…)

L. 103-104 Is there a reason why ants were returned to the nest vicinity in groups, other than convenience of recording? Could this have had an effect on the response to the treated ants, e.g., could the presence of ‘rejected’ ants make resident ants more wary of / aggressive towards any ant outside the nest? I would like to see this briefly discussed in the manuscript.

L. 116 What does LB stand for?

L. 142 Which function in the package lme4 was used (I believe the package includes three different functions)?

L. 149 ‘Benjimini’ should read ‘Benjamini’.

L. 184-188 It might be worth stating here that your antibiotic treatment would have left the ant’s CHC profile unaltered. So while such treatment would have eliminated some familiar odours, others would still be there.

Reviewer 2 ·

Basic reporting

No comments

Experimental design

No comments

Validity of the findings

No comments

Additional comments

Experimental modulation of external microbiome affects nestmate recognition in harvester ants (Pogonomyrmex barbatus)

This is a very nice experimental study where the external microbiome of ants were carefully manipulated to demonstrate that this affects nest mate recognition. I very much like the discussion on the social immunity too.

The manuscript is well written and easy to follow. I have few minor queries/suggestions:
L146: CFUs: Perhaps indicate at the first instance what this acronym stands for. It is in L158, where I found out that this means colony forming units - needs a bit of an explanation.

Can you comment or discuss how the duration of the microbiome on the ant affects whether ants are treated as nest mates or non-nestmates?

Any suggestions on the how the microbiome itself may change the structure of CHC?

---

## Round 0.2 · accepted · Accept

I am happy that you have considered all the comments and made changes where appropriate. Manuscript is now ready for publication.

Nice work,
Regards
Nigel